# Hydrothermal Ageing Effect on Reinforcement Efficiency of Nanofibrillated Cellulose/Biobased Poly(butylene succinate) Composites

**DOI:** 10.3390/polym14020221

**Published:** 2022-01-06

**Authors:** Olesja Starkova, Oskars Platnieks, Alisa Sabalina, Sergejs Gaidukovs

**Affiliations:** 1Institute for Mechanics of Materials, University of Latvia, Jelgavas 3, LV-1004 Riga, Latvia; alisasabalina@gmail.com; 2Faculty of Materials Science and Applied Chemistry, Institute of Polymer Materials, Riga Technical University, P. Valdena 3/7, LV-1048 Riga, Latvia; oplatnieks@gmail.com (O.P.); sergejs.gaidukovs@rtu.lv (S.G.)

**Keywords:** biodegradable polymer, biocomposite, water diffusion, modeling, environmental ageing, durability, mechanical properties, thermomechanical properties, adhesion parameter

## Abstract

Nanofibrillated cellulose (NFC) is a sustainable functional nanomaterial known for its high strength, stiffness, and biocompatibility. It has become a key building block for the next-generation of lightweight, advanced materials for applications such as consumer products, biomedical, energy storage, coatings, construction, and automotive. Tunable and predictable durability under environmental impact is required for high performance applications. Bio-based poly (butylene succinate) (PBS) composites containing up to 50% NFC content were designed and aged in distilled water or at high relative humidity (RH98%). PBS/NFC composites are characterized by up to 10-fold increased water absorption capacity and diffusivity and the data are correlated with model calculations. Aged samples exhibited decreased crystallinity and melting temperature. Incorporation of NFC into PBS showed up to a 2.6-fold enhancement of the elastic modulus, although accompanied by a loss of strength by 40% and 8-fold reduction in the strain at failure of maximally loaded composites. Hydrothermal ageing had almost no influence on the tensile characteristics of PBS; however, there were considerable degradation effects in PBS/NFC composites. Altered reinforcement efficiency is manifested through a 3.7-fold decreased effective elastic moduli of NFC determined by applying the Halpin–Tsai model and a proportional reduction of the storage moduli of composites. The adhesion efficiency in composites was reduced by hydrothermal ageing, as measured Puckanszky’s adhesion parameter for the strength, which decreased from 3 to 0.8. For the loss factor, Kubat’s adhesion parameter was increased by an order. PBS filled with 20 wt.% NFC is identified as the most efficient composition, for which negative environmental degradation effects are counterbalanced with the positive reinforcement effect. The PBS matrix can be used to protect the NFC network from water.

## 1. Introduction

Preserving our planet requires sustainable technologies with low environmental impact to produce high-value products. This has attracted research interest to abundant and renewable materials. Cellulose meets all the criteria, being by far the most abundant renewable resource in the biosphere, having exceptionally tunable morphology, and high mechanical properties [1,2]. Advanced material structure modeling and preparation requires the application of nanocellulose. Nanofibrillated cellulose (NFC) is a sustainable functional nanomaterial known for its high strength, stiffness, and biocompatibility. It has become a key building block for the next generation of lightweight, advanced materials for applications such as consumer products, biomedical, energy storage, functional and decorative coatings, construction, and automotive [3,4]. NFC can form rigid mechanical systems with unique structure that can accommodate various modifications. Combining NFC with polymer into the nanocomposite results in additional layer of protection and functionality from matrix [5,6]. The polymer selection highly benefits bio-based and biodegradable materials to expand upon sustainability and biomedical applications.

Poly (butylene succinate) (PBS) is a relatively new bio-based and biodegradable polyester with high dimensional stability and low water absorption during immersion, even at elevated temperatures [7,8]. This low water sensitivity is one of the key advantages for selecting PBS over other popular biopolyesters such as poly (lactic acid) and polyhydroxyalkanoates. PBS potential applications, such as biomedical, food packaging, agriculture, and automotive, have been evaluated in literature [9,10], showing promising results. PBS biodegradability has been assessed in various studies, and bio-based fillers such as cellulose slightly promote decomposition by promoting water absorption [11,12,13]. As a result, a composite with a higher loading of hydrophilic fillers has better biodegradation capabilities. It has also been demonstrated that PBS remains stable for everyday use items and only starts to degrade when buried in the soil [7]. The mechanisms of biodegradation and some methods for predicting the “end of life” of various bioplastics are discussed in [14]. 

Owing to excellent intrinsic properties and 2D geometry, NFC particles are considered as promising eco-friendly reinforcement fillers for polymer composites [1,15], aerogels and foams [16,17]. Incorporation of NFC into various biopolymers allows material designers to develop advanced sustainable yet durable bioplastics with tailored mechanical performance and biodegradability [5,7]. Depending on material applications and related requirements for specific properties, the amount of cellulose fillers in biopolymer-based composites can vary from a few percent up to 50–70% [18,19,20,21]. The reinforcement efficiency of NFC on composite stiffness normally increases with the higher content of the filler, although this improvement is often accompanied by material embrittlement and loss of adhesion efficiency at high loadings. This applies if NFC does not agglomerate. Coupling agents are used for addressing these shortcomings in order to improve stress transfer at the polymer/filler interface [8,21]. By achieving reasonable compatibility between the composite constituents and their effective dispersion, highly loaded composites with uniquely high stiffness can be developed for advanced applications, e.g., wood–plastic composites [21]. 

Various strategies have been applied in nanocomposite preparation, i.e., melt mixing, solvent casting, and more advanced approaches such as in situ polymerization [22,23]. Direct melt mixing has been shown to be ineffective in the case of NFC, resulting in poor dispersion, damage to the nanostructure of the filler, and even thermal oxidation [24,25]. Thus, it is generally recommended to use solvent processing techniques for the preparation of NFC composites [26]. Solvent casting is divided depending on polymer matrix solubility, i.e., water soluble or non-water soluble. In the case of PBS, solvent exchange for NFC can be applied to prepare the nanocomposite. A different approach is to use the masterbatch method, by preparing a highly loaded system that is then redispersed using melt processing methods. One masterbatch method involves using solvent exchange for NFC and another involves using a water-soluble polymer matrix such as poly (ethylene glycol) (PEG) or poly (vinyl alcohol) as a matrix followed by subsequent blending with PBS. The use of PEG/NFC dispersions in hydrophobic polymer matrix has been shown to be an inefficient strategy due to PEG acting like a plasticizer [27]. Further processing of the solvent cast nanocomposite using methods such as extrusion and injection molding can have a significant impact on properties by achieving fibril orientation [28]. While different processing methods can result in large differences in composite performance properties, in this study we focus on composites’ ability to retain those properties in a PBS matrix when exposed to a hydrothermal ageing environment.

Advanced and functional applications impose exceptionally high requirements for materials’ durability. NFC hybrid materials are highly moisture sensitive [6]. Water absorption from the environment (hydrothermal degradation) significantly reduce the durability and service lifetime of these materials. Insufficient long-term performance studies under environmental ageing restrict the usage of biopolymer hybrid systems. Some studies indicate a significant difference in resistance to environmental ageing between conventional non-biodegradable fossil polymers and biopolyesters [29,30]. The durability of biodegradable plastics, including PBS-based nanocomposite systems, is reported in [31,32]. Hydrothermal ageing of highly loaded cellulose-based biopolymer systems is studied in [8,33] demonstrating their particularly high susceptibility to environmental impact. Property degradation is normally related to water absorption capacity and barrier properties of bioplastics [34,35], thus making NFC less competitive in terms of environmental stability compared to hydrophobic materials, e.g., carbon, nanofillers. Development of sustainable biocomposites with tailored durability and biodegradability performance is a major task for the growing circular economy. Some modelling tools for predicting environmental ageing and degradation of polymers and polymer composites are reported in [14]. Despite continuously growing interest in the research of NFC-based biopolymer composites, most studies are focused on the characterization of their basic engineering properties and biodegradability, while environmental impacts on durability performance and the role of NFC in it is rarely reported.

The present study is aimed at quantitative characterization of the hydrothermal impact on the reinforcement and adhesion efficiency of PBS-based composites with NFC loading up to 50 wt.%. Water absorption and diffusivity were studied in composites under humid and warm water environments to model different degradation scenarios. The large disparity in NFC loading allows to identify critical structural transitions in the NFC network and their contribution to the environmental durability of composites. Structural changes and the role of NFC on composite hydrothermal degradation were revealed through calorimetric studies. Mechanical and thermomechanical properties were studied, while changes in the reinforcement and adhesion efficiency of NFC on the tensile stiffness and strength of composites were evaluated through model parameters. The results allow us to identify the most efficient composition, which is characterized by reasonable mechanical performance and hydrothermal stability to facilitate advanced hybrid systems. 

## 2. Materials and Methods

### 2.1. Materials and Samples’ Processing

PTT MCC Biochem Company Ltd. (Bangkok, Thailand) supplied biobased poly (butylene succinate) (PBS) in the form of pellets. BioPBS™ FZ71PB^®^ grade was selected as it provides balanced properties for use in a wide range of applications. This grade is characterized by a 115 °C melting temperature and a 1.26 g/cm^3^ density. Kraft pulp (hardwood birch) was purchased from Metsä Fibre. Pulp was mechanically treated and processed into a nanofibrillated cellulose (NFC) dispersion using a method described in our previous study [20]. The NFC used was from the same batch, as described in [20], with a diameter *d* in the range of 15–50 nm and a length *l* of 200–550 nm. The average aspect ratio for *l*/*d* is assumed to be equal to 10 in the present study. Chloroform and N,N-dimethylformamide (DMF) were purchased from Merck KGaA (Darmstadt, Germany).

Pure NFC films for water absorption studies were produced from NFC gel suspended in water and mixed with a high-shear mixer (2 min, 5500 rpm). The final concentration of suspension was adjusted to 1 wt% NFC. The NFC film was prepared by casting suspension in a polystyrene Petri dish at room temperature (20 °C). Films were dried in ambient conditions until the water evaporated, then in a thermostat at 50 °C for 48 h. The resulting thickness of the NFC films was 8 ± 1 μm. 

### 2.2. Nanocomposite Preparation

PBS granules were dried in a vacuum furnace (J.P. Selecta, Barcelona, Spain) according to the manufacturer’s instructions (60 °C, 8 h). PBS was dissolved in chloroform using magnetic stirring for 1 h. The NFC water suspension was diluted with DMF and solvent replacement was carried out using solvent-assisted centrifugation. The process was carried out three times, and a high-shear mixer L5M-A (Silverson Machines LTD, Waterside, UK) was used for homogenization (2 min, 5500 rpm). The NFC/DMF suspension was mixed with the PBS/chloroform solution using a high-shear mixer (2 min, 5500 rpm). Solution casting was used to obtain films. Dry films were further dried in a vacuum furnace (60 °C, 8 h). The obtained films were then compression molded (Carver CH 4386, Diamond, MO, USA) to thickness of a 0.2 mm (140 °C, preheating 2 min, 3 min compression under 3 MT). Films were cooled between steel plates (30 kg of thermal conductive mass).

Six compositions with different contents of NFC were produced (Table 1). The volume contents of NFC were calculated using the density ρ_cellulose_ = 1.6 g/cm^3^ [21]. Density of composite samples increased with NFC content up to 5% for maximally loaded samples. This increase correlates well with calculations according to the rule of mixture, indicating a reasonable structural homogeneity of the prepared composites. 

Rectangular samples 10 mm × 55 mm were cut from the films for water absorption and tensile tests. Prior to testing, all samples were dried in a thermostat at 50 °C for 4 days until mass stabilization was achieved. Samples were divided into three groups depending on their storage conditions. As-produced samples stored in plastic zipped bags in a desiccator under silica gel at 22 °C were considered the reference samples (denoted as Ref). One group of samples was immersed in distilled water in a glass container and put into a thermostat at 50 °C: these samples are denoted as W50 in the text. The next group of samples were stored in a desiccator under a saturated salt solution of K_2_SO_4_ giving a relative humidity of RH = 98% at 22 °C (denoted as RH98).

### 2.3. Water Absorption Studies 

An amount of absorbed water in samples was determined by gravimetric measurements. The samples were weighed on a regular basis using analytical scales XS205 (±0.01 mg Mettler Toledo, Columbus, OH, USA), and the relative weight changes [%] were calculated as weight gain per weight unit: (1)$$w=\frac{m_{t}-m_{0}}{m_{0}}\times 100$$
where $m_{t}$ is the weight of the immersed sample at time *t*, and $m_{0}$ is the weight of the reference sample. Replicate samples of each composition were taken out of the water by groups, wiped with blotting paper to remove water from the surface, and weighed following the same sequence. Measuring time of each of the group of samples did not exceed five minutes. The mean values were calculated from data of five replicate samples.

### 2.4. Tensile Tests

Uniaxial tensile tests were performed on a Zwick 2.5 kN universal testing machine at a constant crosshead speed of 2 mm/min according to ASTM D882. Tabs made from a paper tape were applied to samples to ensure smooth loading and prevent sample slippage during the test. Five replicate samples for each composition in the reference (Ref) and aged state, i.e., after conditioning in a humid atmosphere for 40 days (RH98) and warm water for 18 days (W50) were tested. Aged samples were tested 5–10 min after their removal from the environment. The secant elastic modulus was determined in the strain range of 0.2–0.5%.

### 2.5. Differential Scanning Calorimetry 

Differential scanning calorimetry (DSC) was performed using the DSC-1 device (Mettler Toledo, Columbus, OH, USA). The measurements were made for samples with an average weight of 10–12 mg under a nitrogen atmosphere, from 25 °C to 150 °C, at a scan rate of ±10 K/min. Melting temperature and crystallinity were evaluated at the first heating scan. The crystalline fraction ${Χ}_{c}$ [%] of PBS/NFC composites was calculated according to the equation:(2)$${Χ}_{c}=\frac{\Delta H}{\Delta H_{PBS}\left(1-\varphi_{f}\right)}\times 100$$
where $\varphi_{f}$ is the weight fraction of NFC filler in the composite, $\Delta H_{PBS}$ is the heat of fusion of the perfect crystal of PBS, and ΔH is the enthalpy of fusion of the studied samples, respectively. The value of $\Delta H_{PBS}$ is 200 J/g [25]. For water saturated samples (RH98 and W50), ΔH in Equation (2) was replaced by ΔH/(1−w) in order to exclude the weight of absorbed water. 

### 2.6. Dynamic Mechanical Thermal Analysis 

Dynamic mechanical thermal analysis (DMTA) was performed in a tension mode on a Mettler Toledo SDTA861e device (Mettler Toledo, Columbus, OH, USA). Tests were carried out on the reference and RH98 samples with dimensions 4 mm × 8.5 mm. Frequency 1 Hz, heating rate of 3 K/min, displacement amplitude of 10 μm, force amplitude 10 N, and a temperature range from −70 to 70 °C measurement parameters were used.

## 3. Results

### 3.1. Water Absorption 

Water absorption curves for RH98 and W50 samples are shown in Figure 1. PBS, as previously reported [8,35,36], has a low water absorption capacity with a total weight gain of less than 1%. Neat polymer and composites possess Fickian-type diffusion behaviour: linear weight gain versus square root of time dependence at the start of water uptake, whereupon saturation is approached. The data are fitted by the Fick’s model assuming 1D through-thickness diffusion [37,38]
(3)$$w\left(t\right)=w_{\infty }\left[1-\frac{2}{\pi^{2}}\overset{\infty }{\underset{m=1}{\sum }}\frac{{\left[1-{\left(-1\right)}^{m}\right]}^{2}}{m^{2}}\exp\left(-{\left(\frac{\pi m}{a}\right)}^{2}Dt\right)\right]$$
where $w_{\infty }$ is the equilibrium water content, *D* is the diffusion coefficient, and *a* is the thickness of the sample. The initial water content at *t* = 0 is assumed to be zero. As seen from Figure 1, the model approximations are in good agreement with the experimental data.

The equilibrium water content of samples increases with the growing content of NFC in the polymer (Figure 2a) following the rule of mixture given by
(4)$$w_{\infty }^{c}=w_{\infty }^{m}\left(1-\varphi_{f}\right)+w_{\infty }^{f}\varphi_{f}$$
where $w_{\infty }^{c}$, $w_{\infty }^{m}$, and $w_{\infty }^{f}$ are the equilibrium water contents of PBS/NFC composite, PBS matrix and NFC filler, respectively. 

Water sorption capacity increases with NFC content and more extensively for W50 samples compared to their RH98 counterparts (up to an 8- and 10-fold increase, respectively). Equation (4) gives reasonable approximations with $w_{\infty }^{f}$(RH98) = 8% and $w_{\infty }^{f}$(W50) = 11%. The former value correlates well with the experimental value for pure NFC film (Figure 3). A noticeable deviation (overshoot) from the predicted line can be noticed for 50NFC-W50 samples. This fact could be related to structural inhomogeneities, e.g., voids due to imperfect adhesion, formed during processing or initiated by hydrothermal degradation of the material and facilitating additional water ingress [39]. 

The diffusion coefficients of neat PBS and composites were determined by fitting the weight gain curves (Figure 1) by Equation (3). The obtained *D* values versus volume content of NFC are shown in Figure 2b. The diffusivity of PBS greatly increased with the addition of hydrophilic NFC, leading to up to a 3- and 10-fold gain of *D* for maximally loaded RH98 and W50 samples, respectively. The rate of diffusion is greatly accelerated at elevated temperatures and the greater with higher NFC content in PBS. *D* of neat PBS is 5.5 (±0.2) × 10^−4^ mm^2^/h and 7.0 (±0.2) × 10^−4^ mm^2^/h for RH98 and W50 samples, respectively. These values agree well with data reported elsewhere [35].

The Halpin–Tsai model, which is among the most widely used models to predict the elastic modulus, heat conductivity, and water diffusivity in heterogeneous multicomponent polymer systems [40,41,42,43], was applied to fit the data in Figure 2b. The Halpin–Tsai model for the diffusivity is given by Equation (5) [44]: (5)$$D_{c}=D_{m}\left(\frac{1+\xi \eta V_{f}}{1-\eta V_{f}}\right);\;\eta =\frac{D_{f}/D_{m}-1}{D_{f}/D_{m}+\xi }$$
where $V_{f}$ is the volume content of the filler, $D_{m}$ and $D_{f}$ are the diffusion coefficients of the matrix and filler, i.e., PBS and NFC, respectively. ξ=2l/d is the structural parameter related to the aspect ratio of the filler. For NFC under the study, ξ = 20 (Section 2.1). Figure 2b shows a reasonable correlation between the experimentally determined and calculated by Equation (5) diffusivities of PBS/NFC composites. The best fitting results are obtained for $D_{f}$ equal to 0.004 (±0.0005) and 0.015 (±0.001) mm^2^/h for RH98 and W50 samples, respectively. The diffusivity of 50NFC-RH98 samples ($V_{f}=44.1\%$) is slightly lower than that predicted by the Halpin–Tsai model and could be related to the high tortuosity effect in a highly loaded polymer. However, this effect is counterbalanced by additional water ingress into voids in W50 samples.

An attempt has also been made to estimate the water absorption characteristics of pure NFC. Weight gain data for NFC films conditioned under RH 98% at 22 °C are shown in Figure 3. As seen from the graph, the experimental data agrees well with the approximation of Fick’s model given by Equation (3). NFC films reached saturation at *w* = 8% and this value is identical to $w_{\infty }^{f}$ determined by Equation (4) for PBS/NFC composites. The diffusivity of pure NFC determined from the Fick’s approximation is very low: 1.0 (±0.5) 10^−5^ mm^2^/h, which is two orders of magnitude lower than $D_{f}$ determined by Equation (5). NFC in the deposited thin films forms a highly oriented structure, acting as an efficient barrier against the diffusion of water molecules and increasing the tortuosity factor contribution [2,35,43]. This effect is greatly diminished in NFC-filled polymer due to the counterbalancing contribution from the hydrophilic nature of cellulose and structural defects such as NFC agglomerates and microcracks at the interface between the filler and the matrix. These imperfections, which originated during the processing of the material or due to environmental impact, contribute to additional water ingress by the capillary mechanism [39]. The weaker the interfacial adhesion, the greater its contribution. 

The diffusion characteristics of PBS/NFC composites correlate well with the results reported in literature for similar cellulose fiber/hydrophobic polymer systems [36,39], comparison of the absolute values is not actually reasonable due to their high sensitivity to the properties of constituents and processing conditions of composites. For example, saturation of polypropylene/cellulose fiber (30 wt.%) composites immersed in water at 23 °C was reached at *w* = 6%, while the diffusivity was estimated as *D* = 0.0032 mm^2^/h (8.84 10^13^ m^2^/s) [39]. Both characteristics increased considerably with temperature and the content of the fibers. In [45], the equilibrium water content of pure NFC was determined at RH90%, room temperature, and it was 16.6%. 

### 3.2. Tensile Properties

Representative stress–strain diagrams for the reference and aged samples of neat PBS and 20NFC samples are shown in Figure 4. Incorporation of NFC into the polymer resulted in enhancement of its elastic properties, although accompanied by embrittlement of the composites (Figure 4a). The tensile properties of PBS were almost unaffected by hygro- and hydrothermal ageing (Figure 4b), whereas noticeable degradation effects are observed for PBS/NFC composites, where NFC content contributed to higher values. Ageing in warm water resulted in more deleterious property degradation compared to conditioning in a humid environment (Figure 4c). 

The ultimate properties of PBS/NFC composites are shown in Figure 5. The strength (σ*) of the reference samples decreases by 40% with the higher content of NFC: from 30.1 ± 3.2 MPa for neat PBS to 18.6 ± 4.5 MPa for 50NFC samples. However, the σ*of PBS with moderate NFC loadings up to 10 wt.% was not influenced by the presence of the filler. The strain at failure (ε*) decreased from 12.5 ± 2.4% to 1.6 ± 0.4% when comparing neat PBS and 50NFC samples, respectively. The observed embrittlement effect could be explained by inefficient stress transfer at the filler-matrix interface due to low compatibility of the constituents, inefficient dispersion, and agglomerates formed during processing of samples [46,47,48,49].

Hydrothermal ageing resulted in even more progressive strength loss of PBS/NFC composites (Figure 5a). By comparing the σ* values of reference and W50 samples, the drop is estimated to be 16% and 68% for 20NFC and 50NFC composites, respectively. The extent of property degradation increases with NFC content, which is partly explained by higher amounts of absorbed water in highly loaded samples (Figure 2a) and their higher structural defectiveness. Interestingly, the strain at failure is only slightly affected by hydrothermal ageing, and the data for the reference, RH98, and W50 samples are within the data scatter range (Figure 5b).

The elastic moduli (*E*) of composites as a function of NFC volume content in the reference and aged states are compared in Figure 6. NFC demonstrated rather high reinforcement efficiency with up to a 2.6-fold increase in the elastic modulus for 50NFC samples. Ageing caused significant stiffness degradation in composites, as well as a loss of the NFC reinforcing effect. 

The Halpin–Tsai model for randomly oriented short cylindrical fibers, was used to model the elastic moduli of PBS/NFC composites [15,34,45]: (6)$$\begin{array}{c} E_{c}=E_{m}\left[\frac{3}{8}\left(\frac{1+\xi \eta_{L}V_{f}}{1-\eta_{L}V_{f}}\right)+\frac{5}{8}\left(\frac{1+2\eta_{T}V_{f}}{1-\eta_{T}V_{f}}\right)\right]\\ \eta_{L}=\frac{E_{f}/E_{m}-1}{E_{f}/E_{m}+\xi };\;\eta_{T}=\frac{E_{f}/E_{m}-1}{E_{f}/E_{m}+2} \end{array}$$
where *E*_c_, *E_m_*, and *E_f_* are the elastic moduli of the composite, matrix (PBS), and filler (NFC); *l* and *d* are the length and diameter of the NFC, respectively. *V_f_* and ξ are the same as in Equation (5). 

Calculations of *E*_c_ by Equation (6) are shown in Figure 6. According to the experimental data, *E_m_* = 0.57 ± 0.02 GPa. *E_f_* is determined by fitting the experimental data. Calculations with *E_f_* = 3.0, 1.7, and 0.8 GPa give the best approximation results for the reference and RH98 and W50 samples, respectively. This method estimates the plasticization effect of absorbed water on the elastic modulus of NFC indirectly, however with certain simplified assumptions on perfect interfacial adhesion. *E_f_*, and thus, the reinforcement efficiency decreases significantly for water-saturated material by 43% and 73% for RH98 and W50 samples, respectively. This reduction correlates well with higher $w_{\infty }^{\mathrm{NFC}}$ values for the latter samples (Equation (4), Figure 2a). 

Literature data on the elastic properties of NFC are very varied according to the source of cellulose and fiber aspect ratio. *E_f_* is found in the range of 15–140 GPa [15,45,50]. Thus, the determined *E_f_* values of NFC under study are highly underestimated compared to literature data. This could be related to the low properties of the produced NFC itself or the limited compatibility of PBS and NFC, resulting in dispersion and adhesion shortcomings. Because both factors are difficult to distinguish, reasonable data comparison within the same compositions is possible. 

For composites, the strength versus the elastic modulus can be plotted to evaluate the correlation between reinforcement and adhesion efficiency. Figure 7 shows the data for the reference and aged PBS/NFC samples. Composites with low NFC content (up to 20 wt.% NFC) exhibit higher stiffness with moderate strength loss compared to neat PBS. The same trend is maintained after hydrothermal ageing. At higher loadings, the positive NFC reinforcing effect is counterbalanced by negative environmental degradation effects on both NFC and stress transfer at the NFC/PBS interface. The dramatic loss of both the mechanical characteristics observed for highly loaded samples makes them practically inapplicable in harsh environments. By correlating two opposite contributions, optimal NFC content could be determined for specific applications and service requirements. 

A decrease in the strength of particulate composites is related to the degree of interfacial adhesion between the polymer matrix and fillers. The adhesion efficiency could be quantitatively assessed by using Pukanszky’s adhesion parameter [41,48,51]. The Pukanszky’s model for strength is given by [51]
(7)$$\sigma_{c}=\left[\frac{1-V_{f}}{1+2.5V_{f}}\sigma_{m}\right]\exp\left(BV_{f}\right)$$
where σ_c_ and σ_m_ are the strengths of the composite and polymer matrix, respectively. *B* is an empirical constant, also known as Pukanszky’s adhesion factor. Parameter *B* is defined as a measure of the load carried by particles; it approaches zero for poorly bonded particles and increases for efficient polymer-particle interface adhesion interactions [48,51]. After some rearranging and applying a natural logarithm, Equation (7) takes the form
(8)$$\ln\sigma_{red}=\ln\left[\frac{1+2.5V_{f}}{1-V_{f}}\sigma_{c}\right]=\ln\sigma_{m}+BV_{f}$$
where $\sigma_{red}$ is the reduced strength. As follows from Equation (8), the plot $\ln\sigma_{red}$ versus $V_{f}$ is a straight line characterized with a slope *B*. 

Figure 8a shows the reduced strength of PBS/NFC as a function of NFC volume content. For comparison, data for other similar composites are also presented. Polyvinylalcohol (PVA) filled with NFC [50] and PBS filled with wine lees (WL) [40]. The adhesion factors *B*, according to Equation (8), are indicated by the slope of the linear dependences. PBS/NFC is characterized by reasonable adhesion efficiency with *B* = 3, which is higher than *B* of PBS/WL (*B* = 1.6) and lower than that of PVA/NFC (*B* = 7.6). The high *B* value for the latter material is most likely related to the good compatibility of hydrophilic PVA with hydrophilic NFC, in contrast to hydrophobic PBS. Similar *B* values in the range of 0.8–5.3 were reported in other studies [8,41,48,51]. 

Hydrothermal ageing results in deterioration of the polymer matrix/filler interface, reducing the strength and adhesion efficiency of composites. The reduced strength as a function of volume content of NFC for the reference, RH98 and W50 samples of PBS/NFC composites is shown in Figure 8b. Aged composites are characterized by $\ln\sigma_{red}vs.V_{f}$ dependence on a lower slope: *B* decreases to 2.4 and 1.4 for RH98 and W50 samples, respectively. A similar 1.7-fold decrease in *B* due to hydrothermal ageing of PBS/microcrystalline cellulose composites was reported in [8]. For other systems, epoxy filled with inorganic fillers, water absorption resulted in a 1.4-fold drop of *B* [52]. 

Noticeable deviations from the linear dependences $\ln\sigma_{red}-V_{f}$ are observed for high NFC loadings $V_{f}\geq 25.2\%$ (30 wt.%) (Figure 8b). The experimental data show highly underestimated values, indicating a poor adhesion efficiency of 30NFC and particularly 50NFC composites. The deviations are the most extensive for W50 aged samples (Figure 8b), which explain the dramatic loss of interfacial adhesion between PBS and NFC. 

### 3.3. Calorimetric Properties

Representative DSC first heating scans for PBS/NFC reference and aged samples are shown in Figure 9. The calorimetric characteristics are listed in Table 2. The DSC curves of PBS often exhibit two endothermic peaks and a small exothermic peak [53]. This has been explained by the recrystallization process, where two competing processes occur at the same time-melting and recrystallization [20,27,54]. The first melting occurs for crystallites that have a defective (incomplete) structure, followed by a recrystallization process into a more perfect structure [55]. Thus, the observed process depends on how the polymer achieves crystallization. In our case, sample films were obtained with the rapid cooling method. Thus, all melting peaks are relatively broad, and perfect crystalline phase was not achieved. Exposure to water contributed to the appearance of an exothermic crystallization peak in all RH98 and W50 samples. 

NFC nucleates the formation of spherulites, which contributes to additional disruption of the PBS crystalline structure [56]. This effect is visible for 5NFC and 20NFC reference samples, where a pronounced shoulder from the low endothermic peak appears. However, in the case of 50NFC, the high NFC loading results in PBS acting more as an interphase modifier than a continuous polymer matrix. Absorbed water interfered with the recrystallization process, thus the melting peak split in two endothermic peaks. Nanocellulose has been shown to form dense percolation network after reaching critical concentration, which is around few percent loading depending on aspect ratio of filler [57,58]. Thus, the initial formation of spherulites is promoted by the NFC filler, but high loading restricts polymer chain arrangement. The high quality of NFC filler contributes to a relatively low impact on the crystallinity of PBS even at high loadings. 

Reduced recrystallization decreased melting enthalpy values. This is visible as a significant drop in crystallinity for aged compositions (Table 2). The highest χ reduction was for RH98 samples with a decrease around 17% for all compositions except 50 NFC. 5NFC samples showed almost identical changes to PBS. Similar observations on the reduced crystallinity caused by hydrothermal ageing of PBS-based wood plastics are reported in [36]. Furthermore, all W50 samples showed a significant melting peak shift to lower temperatures by 3–7 °C. However, no specific trend related to NFC and water content in composites are noticed. As discussed in Section 3.1, W50 samples showed about a 2-fold higher water uptake than RH98. In this case, water significantly penetrated the structure of the nanocomposite. When small water molecules get inside a polymer structure, they reduce intermolecular bonding between macromolecule chains. However, because NFC promoted this process, an even greater amount of water was absorbed. This results in swelling, which induces internal stresses in the structure and promotes the breakdown of intermolecular bonds. Reduced intermolecular bonding then promotes the breakdown of the crystalline structure during melting. Absorbed water reduced intermolecular bonding and interfered with the recrystallization process, shifting PBS/NFC composite melting process to a lower temperature. 

### 3.4. Thermomechanical Properties

The addition of NFC to PBS greatly improved its thermomechanical characteristics. Similar results on DMTA analysis for unaged PBS/NFC prepared by solution casting and melt processing methods are presented in a previous study [20]. Temperature dependences of the storage modulus (*E*′) and loss factor (tan δ) of RH98 samples are shown in Figure 10. Aged composites demonstrated principally similar thermomechanical behavior to the reference samples. Absorbed water caused small changes in the glass transition temperature (*T*_g_), which were in the 2–5 °C range with the maximally loaded sample having the greatest impact. *T*_g_ of the reference samples determined by tan δ peaks are −16.2 °C for neat PBS and −17.8, −17.0, −16.6, −17.3, and −20.2 °C for 5NFC, 10NFC, 20NFC, and 50NFC samples, respectively. 

The storage moduli of PBS, in both the glassy and rubbery regions, increased with the higher content of NFC. The reinforcement efficiency of NFC is compared in Figure 11 determined at different temperatures: −70 °C (glassy state), 22 °C (room or service temperature that corresponds to the glass-to-rubbery transition region of PBS), and 70 °C (rubbery state). The slope of the linear dependences $E_{c}^{\prime }/E_{m}^{\prime }$ (*c* is composite, *m* is matrix) is related to the reinforcement efficiency factor: the higher the slope, greater the filler contribution to the stiffness improvement of the polymer matrix [20,38]. Increased crystallinity of composites could also contribute to the *E*′ increase [36,41]. Although this is not the case for this study, since the crystallinity of PBS is only slightly and negatively affected by NFC (Table 2). The rubbery moduli are, to a great extent, correlated with the degree of interaction between the polymer and NFC, while the reinforcing effect of the filler is eliminated. Thus, an increasing $E_{c}^{\prime }/E_{m}^{\prime }$ dependence for 70 °C (up to 60% improvement for 50NFC), although with a much lower slope than that for −70 °C, indicates a reasonably high interaction degree at the PBS-NFC interface. Almost a 3-fold increase in the rubbery *E*′ (70 °C) is obtained with the addition of NFC. Identical improvements of *E*′ are also observed at 22 °C, i.e., in the transition region. 

The glassy and rubbery storage moduli of PBS/NFC were only slightly affected by ageing in a humid environment, resulting in minor decrease in the slopes of the dependences $E_{c}^{\prime }/E_{m}^{\prime }$ versus NFC content for both characteristic temperatures (Figure 11, shown by errors). At the same time, a great decrease in *E*′ of aged samples is observed at 22 °C. This could be explained by the higher plasticization effect of absorbed moisture in this temperature range. When the temperature is too low (−70 °C) water molecules are inactive and do not contribute to molecular relaxations of the polymer, while under elevated temperatures (70 °C) the plasticizing water effect is negated by extensive structural rearrangements of PBS. $E_{c}^{\prime }/E_{m}^{\prime }$ decrease at 22 °C is finely correlated with tensile test data (Figure 6). It should also be noted that the service temperatures of these types of composites are more likely to be in the range of 20 ± 10 °C, so testing material properties at conditions that are similar to those found in real-world applications is critical. 

The addition of NFC resulted in an enhancement of the damping properties of PBS that appeared in the decreased tan *δ* peaks [20]. Similar improvements are reported elsewhere for different types of composites [25,59]. Absorbed moisture, acting as an efficient plasticizer, further contributes to increased energy dissipation and the lowering and widening of the loss factor [38,60]. Assessment of the tan *δ* of composites also provides valuable data on interfacial adhesion between the polymer matrix and filler. The Kubat’s adhesion parameter is among these quantitative measures of the adhesion efficiency [47,59]. 

For rigid inclusions, the Kubat’s adhesion parameter *A* is given as follows [61]:(9)$$A=\left[\frac{1}{1-V_{f}}\times \frac{\tan\delta_{c}}{\tan\delta_{m}}\right]-1$$
where $\tan\delta_{c}$ and $\tan\delta_{m}$ are the loss factors of the composite and polymer matrix, respectively. The lower *A* values are characteristic of stronger interfacial adhesion between the polymer and filler since the ideal interface does not contribute to dampening. The increase of *A* results from the enhanced energy dissipation and damping ability of the composite related to the polymer’s viscoelastic nature and structural defects, e.g., voids that appear due to poor interfacial interactions between the matrix and filler [47]. 

The adhesion factors of PBS/NFC increased with the higher content of NFC and the more significantly for aged samples. *A* values are in the range of 0.006–0.1 for reference and 0.06–0.8 for aged PBS/NFC samples. Adhesion efficiency is strongly correlated with the mechanical performance of composites. The strength and elastic modulus of PBS/NFC as functions of the adhesion factor *A* are shown in Figure 12. The strength decreases with increasing *A*, while an opposite, almost mirroring, trend is observed for the elastic modulus. 

The greatest differences between the *A* values for reference and aged samples are observed for 30NFC and 50NFC compositions. In other words, the inherently low adhesion efficiency of highly loaded composites makes them greatly susceptible to environmental impacts. Similar observations are made by comparing the Pukanszky’s adhesion parameters *B* (Figure 8b, Section 3.2). The stiffness of composites is mainly related to the reinforcement efficiency of the inclusions, rather than interfacial adhesion between the constituents. NFC single particles and their agglomerates form a rigid network in PBS, resulting in an *E* increase but, at the same time, promoting crack propagation and lower strength [47]. 

## 4. Conclusions

Fully biobased PBS/NFC composites have been designed with high NFC content (up to 50 wt.%) and reasonable reinforcement and adhesion efficiency are retained after hydrothermal ageing. Ageing in a humid RH98% environment at 22 °C and in water at 50 °C were studied to mimic different degradation scenarios.

Water absorption behaviour of all composites follows the Fick’s model, while water absorption capacity and diffusivity increase considerably with NFC loading. Up to a 10-fold increase in the equilibrium water content and diffusion coefficient is obtained for maximally loaded PBS/NFC samples immersed in warm water. The barrier properties of the composite constituents were estimated by applying the rule of mixture and the Halpin–Tsai model. A reasonable correlation between the calculated and experimental values of NFC water saturation content is obtained, while the diffusion coefficients determined by the micromechanical model fitting for composites are considerably higher compared to the experimental values determined for pure NFC films.

The structure of pristine PBS was only slightly affected by NFC addition, with no extensive differences in its crystallinity (except for 50NFC composition) and melting temperature. Both thermal characteristics decreased due to absorbed water interference, although less extensively in the case of 50NFC. 

Incorporation of NFC into PBS resulted in up to 2.6-fold enhancement of the elastic modulus, while accompanied by a loss of strength by 40% and an 8-fold reduction in the strain at failure of maximally loaded composites. Tensile properties of PBS were almost not affected by hydrothermal ageing, while noticeable degradation effects were observed for PBS/NFC composites. A greater decrease in the mechanical characteristics was observed for compositions with higher NFC content. This fact is related to the higher amount of absorbed water in composites. The reinforcement efficiency of NFC on the stiffness of PBS decreases significantly after hydrothermal ageing. The effective elastic modulus of NFC in composites was reduced from 3 GPa for reference samples to 1.7 and 0.8 GPa for hygro- and hydro-thermally aged samples, respectively. Calculated from the Halpin–Tsai model. The adhesion efficiency in PBS/NFC was evaluated through the Puckanszky adhesion parameter. *B* = 3 was found for the reference PBS/NFC, while water ageing resulted in its 2-fold reduction. 

Absorbed water resulted in moderate plasticization effects in composites, which appeared in a shift of *T*_g_ for 2–5 °C and a decrease of the glassy storage moduli by 10%. Ageing effects on the reinforcement efficiency on the storage moduli of PBS/NFC composites were evaluated at three characteristic temperatures corresponding to the glassy, rubbery, and glass-to-rubbery transition. The greatest changes in the composite storage moduli related to those of the matrix are revealed at an ambient temperature corresponding to the transition region and service temperature range. Three-fold moduli improvements are obtained for the reference samples, while the reinforcement efficiency decreased by a factor of 2 after the conditioning of samples in a humid environment. The damping properties of PBS were altered by NFC addition and environmental ageing. The Kubat’s parameter, used as a quantitative measure of the adhesion efficiency of composites, was in the range of 0.06–0.1 for the reference samples and increased by almost an order due to degradation. These findings are consistent with the decrease in tensile strength and the decrease in the Puckanszky adhesion parameter.

The results will contribute to the development of advanced sustainable biocomposites with excellent biodegradability and durability performance, extending their applications and thus benefiting the growth of the circular economy.

## Figures and Tables

**Figure 1 polymers-14-00221-f001:**
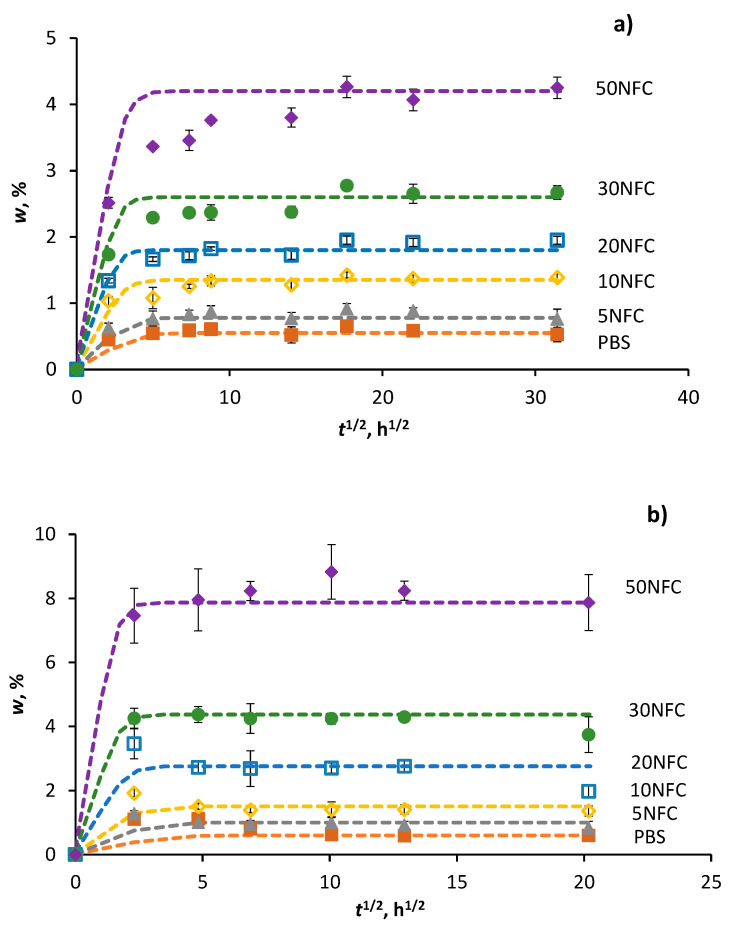
Weight gain of PBS/NFC composites conditioned under RH 98%, 22 °C (**a**) and in water at 50 °C (**b**). Lines are calculated by Equation (3).

**Figure 2 polymers-14-00221-f002:**
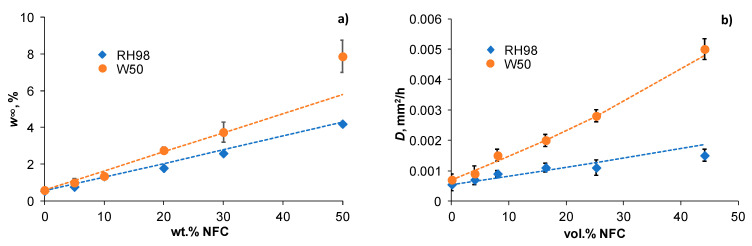
Equilibrium water content of PBS/NFC composites as a function of NFC weight content. Lines are approximations by Equation (4) (**a**); Diffusivity of PBS/NFC composites as a function of NFC volume content. Lines are approximations by Equation (5) (**b**).

**Figure 3 polymers-14-00221-f003:**
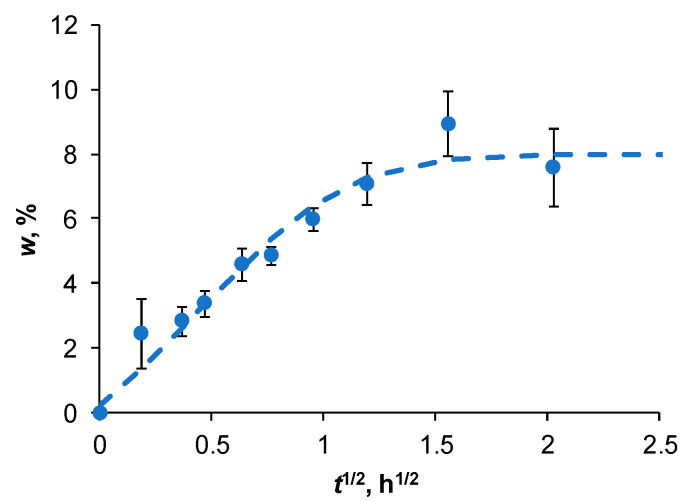
Weight gain of pure NFC film under RH 98%, 22 °C. Line is approximation by the Fick’s model, Equation (3).

**Figure 4 polymers-14-00221-f004:**
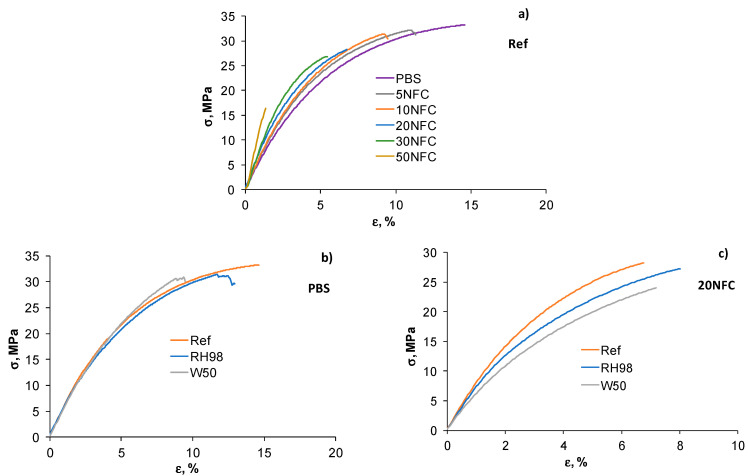
Stress–strain diagrams for reference PBS/NFC composites (**a**) and aged PBS (**b**) and 20NFC (**c**) samples.

**Figure 5 polymers-14-00221-f005:**
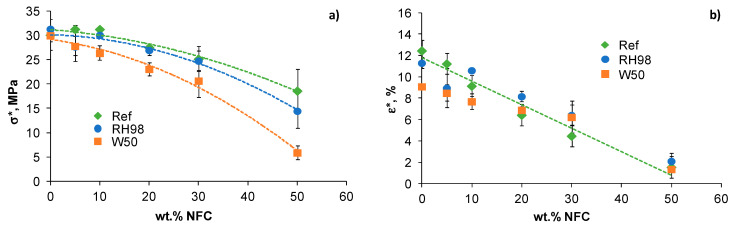
Strength (**a**) and strain at failure (**b**) of PBS/NFC composites as functions of NFC weight content. Lines are approximations.

**Figure 6 polymers-14-00221-f006:**
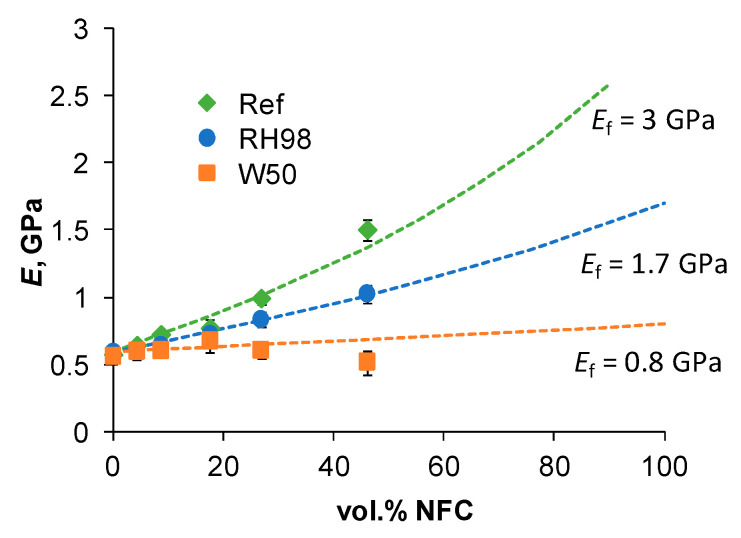
Elastic modulus as a function of NFC volume content for reference and aged samples. Lines are approximations by Equation (6) with different *E_f_*.

**Figure 7 polymers-14-00221-f007:**
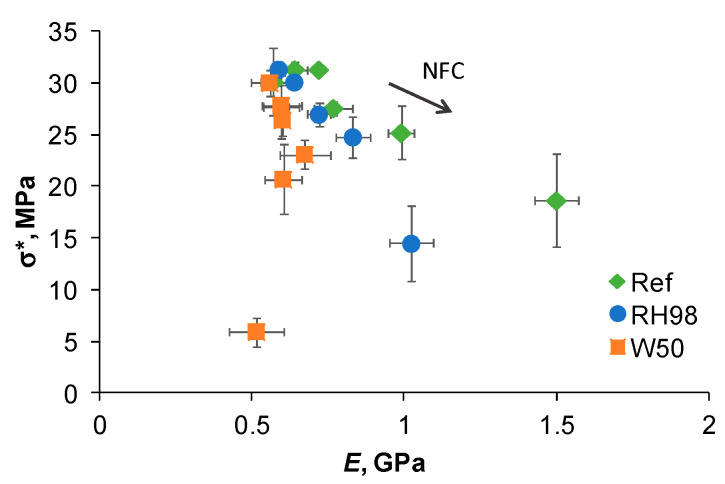
Strength versus elastic modulus of PBS/NFC reference and aged samples.

**Figure 8 polymers-14-00221-f008:**
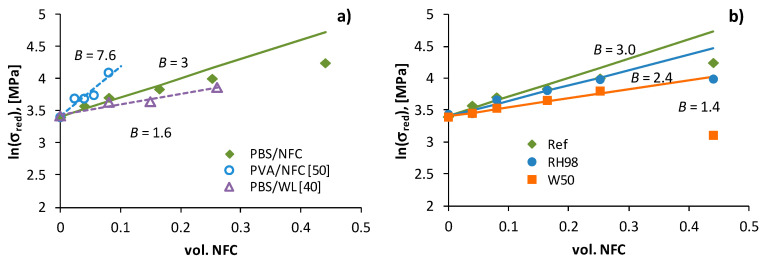
Reduced strength of composites as volume content of the filler for reference PBS/NFC, PVA/NFC [50] and PBS/WL [40] (**a**); and aged PBS/NFC (**b**). Lines are linear approximations with slope *B*.

**Figure 9 polymers-14-00221-f009:**
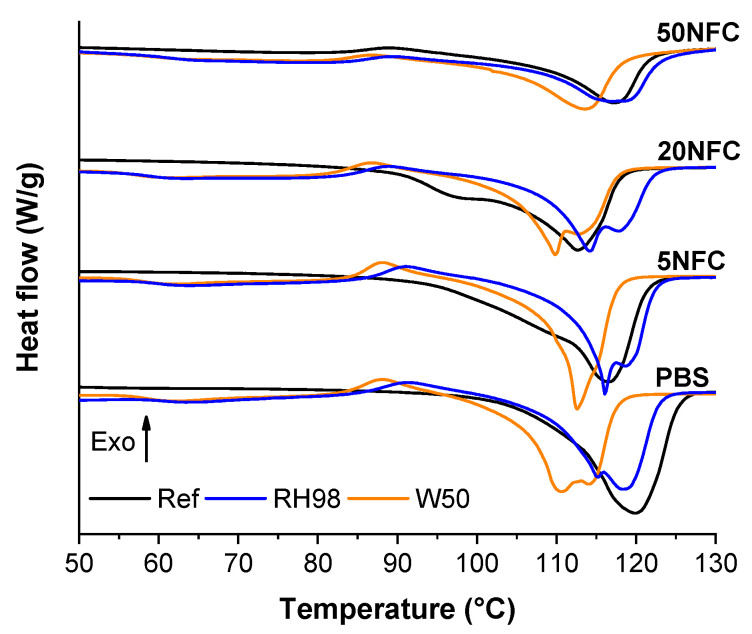
DSC first heating curves for reference and aged PBS/NFC samples.

**Figure 10 polymers-14-00221-f010:**
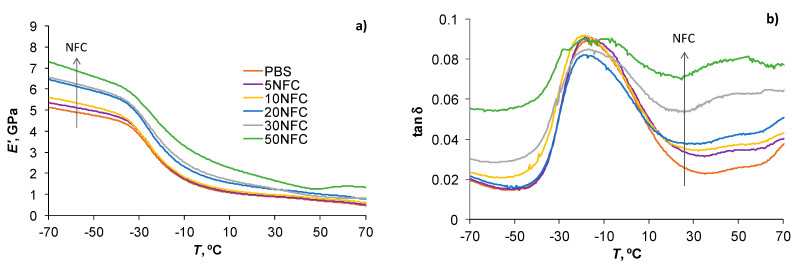
Storage modulus (**a**) and tan δ curves (**b**) for PBS/NFC samples aged under RH98%.

**Figure 11 polymers-14-00221-f011:**
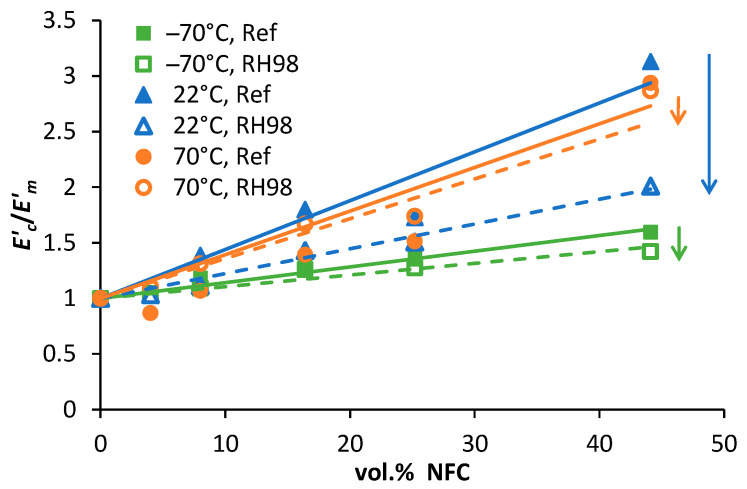
Relative storage moduli as a function of volume content of NFC of the reference and RH98 samples determined at −70, 22, and 70 °C. Solid and dotted lines are approximations.

**Figure 12 polymers-14-00221-f012:**
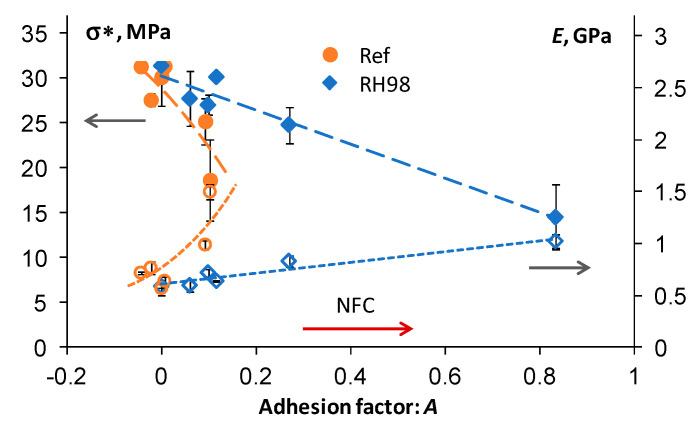
Strength (filled symbols) and elastic modulus (open symbols) as functions of the adhesion factor *A* for reference and RH98 PBS/NFC samples. Lines are approximations.

**Table 1 polymers-14-00221-t001:** Prepared PBS/NFC composites and their densities.

Sample	NFC wt.%	NFC vol.%	ρ ^1^, g/cm^3^
PBS	0	0	1.260 ± 0.0018
5NFC	5	4.8	1.266 ± 0.0025
10NFC	10	8.0	1.273 ± 0.0082
20NFC	20	16.4	1.296 ± 0.0039
30NFC	30	25.2	1.307 ± 0.0164
50NFC	50	44.1	1.326 ± 0.0072

^1^ determined experimentally by hydrostatic weighing in isopropanol (ρ_iso_ = 0.786 g/cm^3^).

**Table 2 polymers-14-00221-t002:** Calorimetric characteristics of PBS/NFC for reference and aged samples (1st heating scan).

Sample	χ (%)			*T*_m_ (°C)		
	Ref	RH98	W50	Ref	RH98	W50
PBS	54.6	38.0	37.6	118	118	111
5NFC	54.7	36.8	37.5	116	116	113
10NFC	49.6	36.7	34.4	115	114	111
20NFC	50.5	33.9	35.8	113	114	110
30NFC	53.6	34.3	36.3	114	115	110
50NFC	43.1	32.4	39.7	117	117	114

## Data Availability

The data presented in this study are openly available in Mendeley Data at DOI:10.17632/8pnvjtcbwc.1, 2021.

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
