# Peer review of "Hydrothermal Ageing Effect on Reinforcement Efficiency of Nanofibrillated Cellulose/Biobased Poly(butylene succinate) Composites"

_polymers, 2022, doi:10.3390/polym14020221_

Round 1
Reviewer 1 Report
Dear authors,
The presented manuscript describes a route to prepare reinforced biobased polybutylene succinate (PBS) composites by reinforcing them with nanofibrillated cellulose (NFC).
The paper is well structured and provides relevant details of the preparation of the composites and their complete characterization focusing on their ageing properties which are not widely explored.
The introduction briefly presents the motivation of the manuscript and the use of nanofillers to modify the properties of biopolymers. However, it would be beneficial for the readers to provide more references of previous works and strategies to improve the properties of biopolymers. If the authors consider it relevant focus mainly on PBS, but more details are expected on different strategies to incorporate nanofillers such as NFC. Modification strategies and relevant results are necessary to improve the introduction and fully present the motivation of the present work. Finally, potential applications of these materials and their processing technologies are expected. These materials are processed by technologies such as blown moulding or extrusion casting that should be at least mentioned in the present manuscript as those technologies may highly affect the final properties of the composites.
Materials and characterization techniques are well presented and structured. Although, it would be necessary to compare the results obtained by solvent casting with other technologies such as extrusion, casting and blowing in order to further evaluate the conclusions achieved in the present manuscript.
Mechanical and thermomechanical analysis are well structured, presented and justified. The observations are well justified and discussed, the figures are representative of the presented results.
The authors mention that the work will contribute to the development of biocomposites with tailored biodegradability and no result is presented related to this statement. Please comment on that issue or provide references that justify it.
Considering all these points my recommendation to the editor will be to reject the manuscript and reconsider it after these modifications are included.
Author Response
Dear authors,
The presented manuscript describes a route to prepare reinforced biobased polybutylene succinate (PBS) composites by reinforcing them with nanofibrillated cellulose (NFC).
The paper is well structured and provides relevant details of the preparation of the composites and their complete characterization focusing on their ageing properties which are not widely explored.
The introduction briefly presents the motivation of the manuscript and the use of nanofillers to modify the properties of biopolymers. However, it would be beneficial for the readers to provide more references of previous works and strategies to improve the properties of biopolymers. If the authors consider it relevant focus mainly on PBS, but more details are expected on different strategies to incorporate nanofillers such as NFC. Modification strategies and relevant results are necessary to improve the introduction and fully present the motivation of the present work. Finally, potential applications of these materials and their processing technologies are expected. These materials are processed by technologies such as blown moulding or extrusion casting that should be at least mentioned in the present manuscript as those technologies may highly affect the final properties of the composites.
- Dear Reviewer, Thank you for your comments and suggestions! The introduction section has been updated to reflect our choice for the composite preparation and give a brief overview on the impact of various methods. Relevant references are added.
Materials and characterization techniques are well presented and structured. Although, it would be necessary to compare the results obtained by solvent casting with other technologies such as extrusion, casting and blowing in order to further evaluate the conclusions achieved in the present manuscript.
- There are several reviews that deal with comparison of performance properties (e.g., [7], [26]), for this study we focus on relatively unexplored topic that is ability to retain performance properties after a hydrothermal ageing. When more relevant research is available for aged composites, then a comparison can be made.
Mechanical and thermomechanical analysis are well structured, presented and justified. The observations are well justified and discussed, the figures are representative of the presented results.
The authors mention that the work will contribute to the development of biocomposites with tailored biodegradability and no result is presented related to this statement. Please comment on that issue or provide references that justify it.
- We agree that in this research we did not study this topic. The wording has been changed. The introduction was supplemented with information about biodegradation with relevant references to literature.
Considering all these points my recommendation to the editor will be to reject the manuscript and reconsider it after these modifications are included.
Reviewer 2 Report
This manuscript by Olesja Starkova, Oskars Platnieks, Alisa Sabalina, Sergejs Gaidukovs reports Hydrothermal ageing effect on reinforcement efficiency of nanofibrillated cellulose/biobased poly(butylene succinate) composites.
This manuscript is the consequence of the authors' research interests and publications in this topic. The authors are experienced in this field and have displayed domain expertise in a variety of characterization methodologies. Generally, the manuscript is well written and well structured, the experimental design seems to be extremely relevant, and the methodology is adequate.
The authors discuss a hot topic of diverse biopolymers, which enables material designers to create improved sustainable but durable bioplastics with customized mechanical performance and biodegradability. Furthermore, the description and interpretation of the different data analysis are both very informative and instructive.
The authors were cautious, always using more than two tests for each trial in order to do a statistical analysis and add an error bar to each measurement.
I recommend this manuscript for acceptance.
Minor points:
- Figure 11 – Units labels “oC”
Author Response
This manuscript by Olesja Starkova, Oskars Platnieks, Alisa Sabalina, Sergejs Gaidukovs reports Hydrothermal ageing effect on reinforcement efficiency of nanofibrillated cellulose/biobased poly(butylene succinate) composites.
This manuscript is the consequence of the authors' research interests and publications in this topic. The authors are experienced in this field and have displayed domain expertise in a variety of characterization methodologies. Generally, the manuscript is well written and well structured, the experimental design seems to be extremely relevant, and the methodology is adequate.
The authors discuss a hot topic of diverse biopolymers, which enables material designers to create improved sustainable but durable bioplastics with customized mechanical performance and biodegradability. Furthermore, the description and interpretation of the different data analysis are both very informative and instructive.
The authors were cautious, always using more than two tests for each trial in order to do a statistical analysis and add an error bar to each measurement.
I recommend this manuscript for acceptance.
Minor points:
- Figure 11 – Units labels “oC”
Dear Reviewer, Thank you for your comments! The figure is updated.
Reviewer 3 Report
The authors discussed the PBS/NFC composites. It seems that this work is quiet interesting. The reviewer’s comments are as follows,
- Since crystallization kinetics is a key issue in the research of polymer composites, the authors should also discuss the crystallization kinetics of PBS/NFC composites.
- How about the biodegradability of the PBS/NFC composites? The authors should also comment the biodegradability of the PBS/NFC composites.
- The authors need to mark the values on the heat flow axis in Figure 9.
Author Response
The authors discussed the PBS/NFC composites. It seems that this work is quiet interesting. The reviewer’s comments are as follows,
Since crystallization kinetics is a key issue in the research of polymer composites, the authors should also discuss the crystallization kinetics of PBS/NFC composites.
- Dear Reviewer, Thank you for your comments and suggestions! We have supplemented DSC analysis section to better reflect impact of NFC, but generally kinetics is studied for concentrations bellow percolation.
How about the biodegradability of the PBS/NFC composites? The authors should also comment the biodegradability of the PBS/NFC composites.
- The introduction has been supplemented.
The authors need to mark the values on the heat flow axis in Figure 9.
- The Figure 9 axis is chosen as an arbitrary value to reflect the process DSC thermogram. And when graphs are stacked, no scale is usually provided. All relevant values have been provided in Table 2, and enthalpies can be calculated from equation 2.